# Evaluation of Ground and Surface Water Hydrochemistry for Irrigation Suitability in Borneo: Insights from Brunei Darussalam

Siti Lieyana Azffri [1,2,*], Chua Siaw Thong [3], Lee Hoon Lim [3], Md Faizan Ibrahim [4], Mario Schirmer [5,6,7] and Stefan Herwig Gödeke [1,*]

1   Geosciences, Faculty of Science, Universiti Brunei Darussalam, Bandar Seri Begawan BE 1410, Brunei
2   Preston GeoCEM (B) Sendirian Berhad, Bandar Seri Begawan BG 3122, Brunei
3   Chemical Sciences, Faculty of Science, Universiti Brunei Darussalam, Bandar Seri Begawan BE 1410, Brunei; 18b2029@ubd.edu.bn (C.S.T.); leehoon.lim@ubd.edu.bn (L.H.L.)
4   Department of Agriculture and Agrifood, Ministry of Primary Resources and Tourism, Berakas BB 3513, Brunei; faizan.ibrahim@agriculture.gov.bn
5   Eawag, Swiss Federal Institute of Aquatic Science and Technology, 8600 Dübendorf, Switzerland; mario.schirmer@eawag.ch
6   Centre of Hydrogeology and Geothermics, University of Neuchâtel, 2000 Neuchâtel, Switzerland
7   Department of Geology and Geological Engineering, Laval University, Quebec City, QC G1V 0A6, Canada
*   Correspondence: lieyana.azffri@gmail.com (S.L.A.); stefan.godeke@ubd.edu.bn (S.H.G.)

**Abstract:** Water for irrigation use plays a critical role in agriculture via supporting crop growth and maintaining food production worldwide. Irrigation water quality evaluations provide useful information for sustainable water practices in many agricultural regions. In Brunei Darussalam, the quality of irrigation water is still poorly understood. The present study aims to investigate the hydrochemical characteristics of water resources in Brunei Darussalam and evaluate their quality and suitability for irrigation use. A total of fifteen sampling locations were chosen from selected ground and surface water sources found in all four Brunei districts. The water samples' physicochemical properties, including pH, EC, and major cations and anions, were measured and compared with FAO standards. Hydrochemical classification based on the Piper diagram revealed that water mainly belongs to the calcium-, magnesium-, and bicarbonate-type, or Type IV water class. The evaluation of heavy metals (Fe, Zn, Cu, Cr, As, and Cd) showed concentrations within the FAO's permissible limits. In this regard, iron showed the highest concentration among the investigated metals. Established water quality indices such as SAR, Na%, RSC, MAR, KR, PS, and IWQI were used to evaluate and classify the water's suitability for irrigation use. Overall, our findings revealed that almost all of the analysed water samples in the study area have low salinity and sodicity risks. However, only 27% of the samples passed the magnesium hazard assessment, and one sample showed a very poor IWQI result. Thus, additional testing and treatment are recommended for these cases. This study provides valuable insights on water quality for present and future utilisation, aiming to contribute to the protection of water resources in Brunei Darussalam.

**Keywords:** hydrochemistry; water quality; irrigation; tropical region; Brunei Darussalam

## 1. Introduction

Water is essential for all forms of life and is one of the earth's most vital resources. Water is critical for energy and food production, socioeconomic growth, healthy ecosystems, and general human existence [1]. However, water issues persist to be a global concern. From water scarcity and pollution to poor sanitation and waterborne diseases, the challenges vary and are often interlinked [2]. Rapid industrialisation, urbanisation, and intensive agriculture have led to a surge in water contaminations, thus, increasing the demand for clean water supply [3]. A growing population is expected to further increase water

supply demands in many regions [4]. Therefore, recognizing the significance of water and implementing sustainable practices to maintain its availability and quality are crucial for protecting water resources for both the current and future generations [1,4].

Water quality evaluations play an important role in water conservation and preservation [5]. It involves hydrochemical investigations and characterization of various physicochemical parameters such as pH, electrical conductivity, major ions, and other chemical constituents present in the water [6]. Although heavy metals are typically found in only small concentrations, they are commonly tested when pollution is suspected or wastewater is reused [7,8]. The findings from water quality evaluations have been useful for assessing the suitability of water for various purposes such as for domestic use, agriculture, and industry [9,10].

Irrigation water plays a critical role in agriculture through supporting crop growth and food production worldwide [11,12]. However, low water quality and highly mineralised irrigation water can have negative effects; for example, minerals can accumulate in soils and be absorbed by crops, causing plant toxicity and nutrient imbalance, thus, compromising soil fertility and overall crop productivity [13,14]. Furthermore, water may infiltrate through the soil into shallow aquifers, further altering groundwater quality [15].

An irrigation water quality guideline provides a framework for understanding water quality and suitability for optimizing water use, subsequently improving crop productivity, and minimizing environmental impacts [16]. The Food and Agricultural Organization of the United Nations (FAO) standard is the most widely used standard water quality guideline for assessing water for irrigation purposes [17–19]. Other well-established water quality indices utilised for classifying irrigation suitability include the sodium adsorption ratio (SAR), sodium percentage (Na%), residual sodium carbonate (RSC), magnesium adsorption ratio (MAR), Kelley's ratio (KR), and potential salinity (PS) [20–22]. In addition, the irrigation water quality index (IWQI) is one of the most effective tools used to evaluate irrigation water quality through encompassing selected key parameters (physical, chemical, or biological) and representing them in a concise and simplified manner [23,24].

Tropical Brunei Darussalam is blessed with abundant rainfall due to its equatorial climate, which provides a substantial surface water resource [25]. In general, surface water resources in Brunei can be considered safe [26]. However, there have been growing concerns about the water quality, especially for residential, agricultural and industrial waterways [27,28]. Furthermore, significant water abstraction from the Tutong River has caused a decrease in water levels, exposing areas of acid sulphate soils [29,30]. As a result, a correlation between low pH values and increased aluminium levels was observed [29]. An investigation of groundwater quality in the Berakas coastal areas further revealed waters with low pH values and high sulphate levels [31]. Moreover, the country's performance with regards to integrated water resource management (IWRM) was assessed as below average in comparison to other countries in Southeast Asia [32], owing partly to the lack of available information.

Surface water is the main source of water supply for agricultural use in Brunei Darussalam [25,33,34]. However, there has been growing interest in the use of groundwater as an alternative water supply for irrigation, especially in water-scarce and rural areas [35–37]. Furthermore, water quality studies for irrigation use in Brunei are limited. The aim of this study is to evaluate the quality and suitability of ground and surface water resources in Brunei Darussalam for irrigation purposes. For a comprehensive overview of the water resources in the country, this study will incorporate samples from groundwater wells, rivers, lakes, and reservoirs found in all four Brunei districts. Results will be compared with the worldwide standard guideline (FAO). Established water quality indices for irrigation such as SAR, Na%, RSC, MAR, KR, PS, and IWQI will be determined. This study will discuss potential risks of contaminants in irrigation water, aiding future decision-making and implementation of appropriate mitigation measures.

## 2. Materials and Methods

### 2.1. Study Area

2.1.1. Geographic Location and Climate

The study area Brunei Darussalam, or simply Brunei, is located on the northwestern part of the Borneo Island, overlooking the South China Sea (Figure 1). The country has an area of about 5765 km$^2$ and is separated into two enclaves [38]. Most of the country's population, administrative, and economic centres are located in the western enclave (Brunei-Muara, Tutong, and Belait districts), which is characterised by steep lowlands. The sparsely inhabited eastern enclave (Temburong District) is hillier and mainly covered by lush woodlands [38].

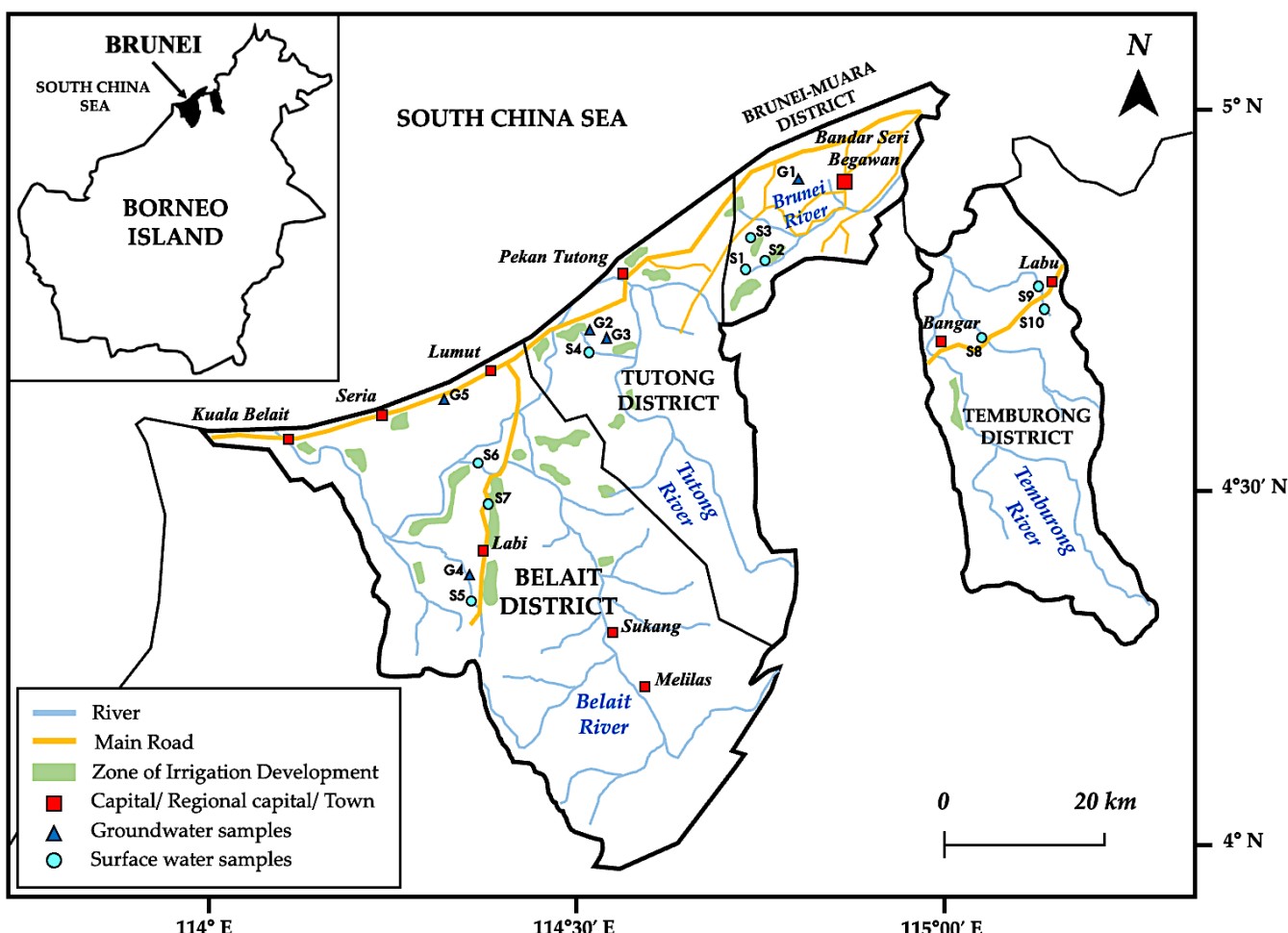

**Figure 1.** Map of Brunei showing the locations of districts, rivers, and sampling points; inset map of Borneo Island showing the location of Brunei. Modified after [25].

Brunei Darussalam is located in the equatorial tropics, with high temperatures and precipitation all year round [39]. Climatic changes in the region are caused by periodic winds of the northeast monsoon, blowing from December to March, and the southeast monsoon, active between June and October. Between 1984 to 2013, the country recorded an average precipitation of about 2976 mm per year, with an annual average temperature of 27.5 °C [40]. There are two seasonal patterns: wet and dry seasons. May to July and October to January are the wet periods, whereas February through March and June through August are the dry periods [39].

2.1.2. Regional Geological and Hydrological Settings

The geology of Brunei Darussalam is made up of thick deltaic sedimentation that overlies deeper and older rock strata. From the Oligocene to the Holocene, rock units can be up to 15 km thick [38,41]. The Neogene rock units with exposures recorded in the onshore areas of Brunei are divided into the Setap, Belait, Lambir, Miri, Seria, and Liang Formations (Figure 2). The Setap Formation comprises deep marine shales. Thick sandstones are found in the Belait Formation, whereas interbedded sands and shales are found in the Seria, Miri, and Lambir Formations. The Liang Formation consists of loosely cemented sands and sandstones with occasional clays and conglomerates. From the Pleistocene to the Holocene, sediments are made up sands and clays [38]. Faults that evolved in deep sedimentary strata are excellent traps for hydrocarbon reserves found in the basin [42].

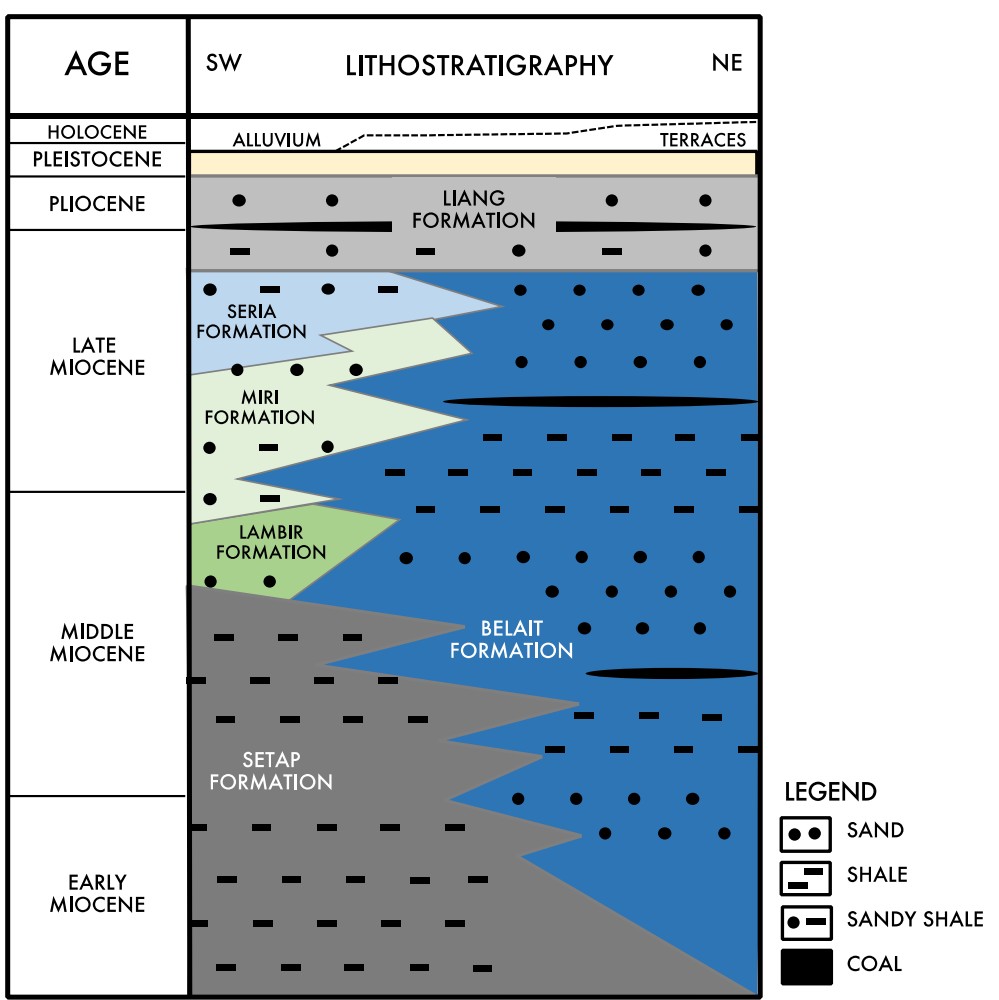

**Figure 2.** Chrono-lithostratigraphy of central onshore Brunei. Modified after [36].

The hydrological setting of Brunei suggests that the country has abundant surface water resources. The current geomorphology of the country was formed during the last significant sea level fall, around 6000 years ago [36]. The four major rivers in Brunei are the Belait River (209 km), Tutong River (137 km), Temburong River (98 km), and Brunei River (41 km) (Figure 1). The Brunei River and the Tutong River serve as the main sources of water supply for domestic, industrial, and agricultural purposes [43]. Additionally, limited groundwater reserves have been identified in the Liang area for potable drinking [25]. Groundwater resources are found mainly within loosely cemented sandstones and fractures along the existing rock formations. The Liang Formation is believed to be the most prolific groundwater formation [43]. Furthermore, the groundwater recharge rate in the country

could be as high as 800 mm/a, with groundwater depths ranging from 10 to 20 m below ground level [44].

## 2.2. Chemical Analysis

A total of five groundwater samples and ten surface water samples were collected in the study area (Figure 1). The coordinates and characteristics of groundwater wells and surface sampling sources are given in Tables 1 and 2. There were no known groundwater wells available for sampling in the Temburong district. The sample coordinates were taken using a digital GPS device (GPSMAP 64S, Garmin, UK). Samples were collected twice in January and in September of 2022. The sample dates were chosen to correspond to the wet season and an increase in irrigation use. Samples were collected and stored in sterile polyethene bottles (500 mL and 1000 mL; unfiltered) and sample vials (15 mL; filtered; 0.45 μm). Filtered samples were treated with nitric acid (0.15 mL). A portable pH/conductivity digital meter (MW801, Milwaukee, USA) was used for pH and EC measurements at the site. The remaining chemical parameters were analysed in the laboratory according to the International Standardization Organization (ISO) and Americal Public Health Association (APHA) [45]. The Jan-2022 samples were analysed at the Eawag Swiss Federal Institute of Aquatic Science and Technology laboratory in Switzerland, whereas the Sept-2022 samples were analysed at the ALS Technichem Sendirian Berhad laboratory in Malaysia. Analytical methods are described in Table 3.

**Table 1.** Coordinates and characteristics of groundwater wells in this study.

| District | Sample ID | Coordinates | | Characteristics |
| | | Latitude | Longitude | |
|---|---|---|---|---|
| Brunei-Muara | G1 | 4°56′06.8″ N | 114°52′49.4″ E | Flowing artesian; irrigation use. Water flow rate of 0.5 $m^3$/h; well depth of 50 m bgl. |
| Tutong | G2 | 4°42′00.6″ N | 114°38′18.6″ E | Flowing artesian; potable water. Water flow rate of 1–2 $m^3$/h; well depth of 20 m bgl. |
| | G3 | 4°41′59.5″ N | 114°38′20.1″ E | Flowing artesian; potable water. Water flow rate of 1–2 $m^3$/h; well depth of 25 m bgl. |
| Belait | G4 | 4°23′07.4″ N | 114°27′11.6″ E | Irrigation use; water flow rate of 5–12 $m^3$/h [37]. Well depth of 80 m bgl; submersible pump depth of 60 m bgl. |
| | G5 | 4°39′07.4″ N | 114°25′32.4″ E | Flowing artesian; potable water. Water flow rate of 28.8 $m^3$/h; well depth of 200 m bgl. |

Note: m bgl: metres below ground level.

**Table 2.** Coordinates and characteristics of surface water sources in this study.

| District | Sample ID | Coordinates | | Characteristics |
| | | Latitude | Longitude | |
|---|---|---|---|---|
| Brunei-Muara | S1 | 4°47′19.9″ N | 114°48′58.7″ E | Wasan river; drainage and source for paddy field irrigation. |
| | S2 | 4°47′38.5″ N | 114°49′17.4″ E | Panchor river; drainage and source for paddy field irrigation. |
| | S3 | 4°48′05.9″ N | 114°48′06.2″ E | Imang reservoir; main source for irrigation in Brunei-Muara. Reservoir capacity of 8 million cubic metres [46]. |
| Tutong | S4 | 4°41′57.6″ N | 114°38′19.3″ E | Penapar river; located near construction site of new dam. |
| Belait | S5 | 4°22′04.3″ N | 114°27′23.6″ E | Rampayoh river; drainage and source for paddy field irrigation. |
| | S6 | 4°32′20.5″ N | 114°28′04.7″ E | Belait river; located near residential area. |
| | S7 | 4°30′56.1″ N | 114°28′28.9″ E | Luagan lake; located near recreational area. |
| Temburong | S8 | 4°43′02.7″ N | 115°06′33.0″ E | Lamaling river; located near residential and commercial areas. |
| | S9 | 4°45′50.2″ N | 115°11′10.8″ E | Labu river; located near recreational area and paddy field. |
| | S10 | 4°45′04.0″ N | 115°12′07.7″ E | Senukoh river; located near recreational area and paddy field. |

**Table 3.** Water quality parameters, analytical methods and detection limits.

| Sampling Event | Parameters | Method | Reference | Detection Limits * |
|---|---|---|---|---|
| January-2022 | Sodium, magnesium, calcium, potassium | Ion chromatography | EN ISO 14911 | 0.5, 1.5 |
| | Chloride, sulphate | Ion chromatography | EN ISO 10304-1 | 0.5, 0.1 |
| | Bicarbonate, carbonate | Titration | EN ISO 9963-1 | 1 |
| September-2022 | Sodium, magnesium, calcium, potassium | Ion chromatography | APHA 3120-N | 0.1 |
| | Chloride | Titration | APHA 4500-Cl-E | 1 |
| | Sulphate | Turbidimetric | APHA 4500-SO-E | 1 |
| | Bicarbonate, carbonate | Titration | APHA 2320B | 1 |
| | Iron, zinc, lead, copper, chromium, cadmium, arsenic | Ion chromatography | APHA 3125B | 0.001, 0.0005 |

Note: * Unit: mg/L.

Charge balance errors (CBE) were calculated using PHREEQC software [47] to ensure the validity of the chemical analysis of the water samples. The ionic charge balance error of all analysed water samples in this study ranged between $\pm10\%$, which is acceptable as per research publication standards [48,49].

## 2.3. Water Classification

Prior to data classification, mean results were calculated for each measured parameter and compared with the FAO standard guideline [16] (Table 4). Water classification for irrigation suitability was conducted using various established water quality indices: SAR [50], Na% [51], RSC [52], MAR [53], KR [54], PS [51], and IWQI [55,56]. The Piper diagram [57] was used for plotting and classifying the water hydrochemistry, while the USSL diagram [58] and Wilcox diagram [59] were used for suitability classification for irrigation use.

**Table 4.** Mean physicochemical properties of ground and surface water samples in Brunei Darussalam based on sampling campaigns in January and September 2022.

| Sample | Sample ID | pH | EC μS/cm | $Na^+$ | $Mg^{2+}$ | $Ca^{2+}$ | $K^+$ | $SO_4^{2-}$ | $Cl^-$ | $HCO_3^-$ |
|---|---|---|---|---|---|---|---|---|---|---|
| | | | | | | | mg/L | | | |
| Groundwater | G1 | 4.8 | 28.5 | 0.7 | 0.2 | 0.4 | 0.1 | 1.5 | 1.9 | 15.5 |
| | G2 | 4.6 | 50.0 | 0.8 | 0.7 | 0.8 | 0.4 | 5.0 | 1.2 | 10.0 |
| | G3 | 5.2 | 68.5 | 0.9 | 2.7 | 1.5 | 0.5 | 17.5 | 1.2 | 28.0 |
| | G4 | 6.3 | 1127 | 190 | 7.4 | 2.1 | 2.8 | 1.0 | 306 | 198 |
| | G5 | 6.5 | 85.0 | 3.5 | 2.3 | 4.3 | 1.3 | 9.0 | 1.0 | 36.0 |
| Surface Water | S1 | 6.1 | 90.5 | 5.3 | 3.4 | 2.5 | 1.7 | 17.5 | 4.0 | 44.0 |
| | S2 | 6.6 | 133 | 6.7 | 4.3 | 5.5 | 2.4 | 15.5 | 5.9 | 53.5 |
| | S3 | 6.9 | 51.0 | 2.8 | 2.1 | 1.1 | 0.9 | 5.0 | 0.8 | 19.5 |
| | S4 | 4.6 | 51.5 | 0.6 | 1.1 | 1.1 | 0.3 | 9.0 | 1.2 | 11.0 |
| | S5 | 5.4 | 38.0 | 0.7 | 1.3 | 0.8 | 0.6 | 8.0 | 0.9 | 28.0 |
| | S6 | 4.9 | 30.0 | 0.1 | 0.3 | 0.5 | 0.1 | 3.0 | 0.5 | 73.0 |
| | S7 | 4.8 | 39.0 | 0.7 | 0.2 | 0.7 | 1.3 | 1.0 | 1.0 | 5.0 |
| | S8 | 6.8 | 32.0 | 1.4 | 1.4 | 1.2 | 0.2 | 3.5 | 1.0 | 36.5 |
| | S9 | 6.3 | 202 | 27.0 | 3.1 | 0.9 | 1.1 | 5.5 | 46.1 | 47.0 |
| | S10 | 6.8 | 51.5 | 4.0 | 1.2 | 0.8 | 0.5 | 3.7 | 2.8 | 34.5 |
| Worldwide Standard Limits | | | | | | | | | | |
| FAO * | | 6.5–8.4 | <750 | <920 | <60 | <400 | <30 | <960 | <1050 | <150 |

Note: * Standard permissible limits are those provided by the Food and Agriculture Organisation of the United Nations (FAO) [16].

## 3. Results and Discussion

### 3.1. General Characteristics of Water Quality

3.1.1. Chemical Composition

General water quality characteristics of ground and surface water samples in Brunei Darussalam are presented in this section. The mean results of measured pH, EC, and major cations (sodium, magnesium, calcium, and potassium) and anions (sulphate, chloride, and bicarbonate) are listed in Table 4 and plotted in the boxplots shown in Figure 3.

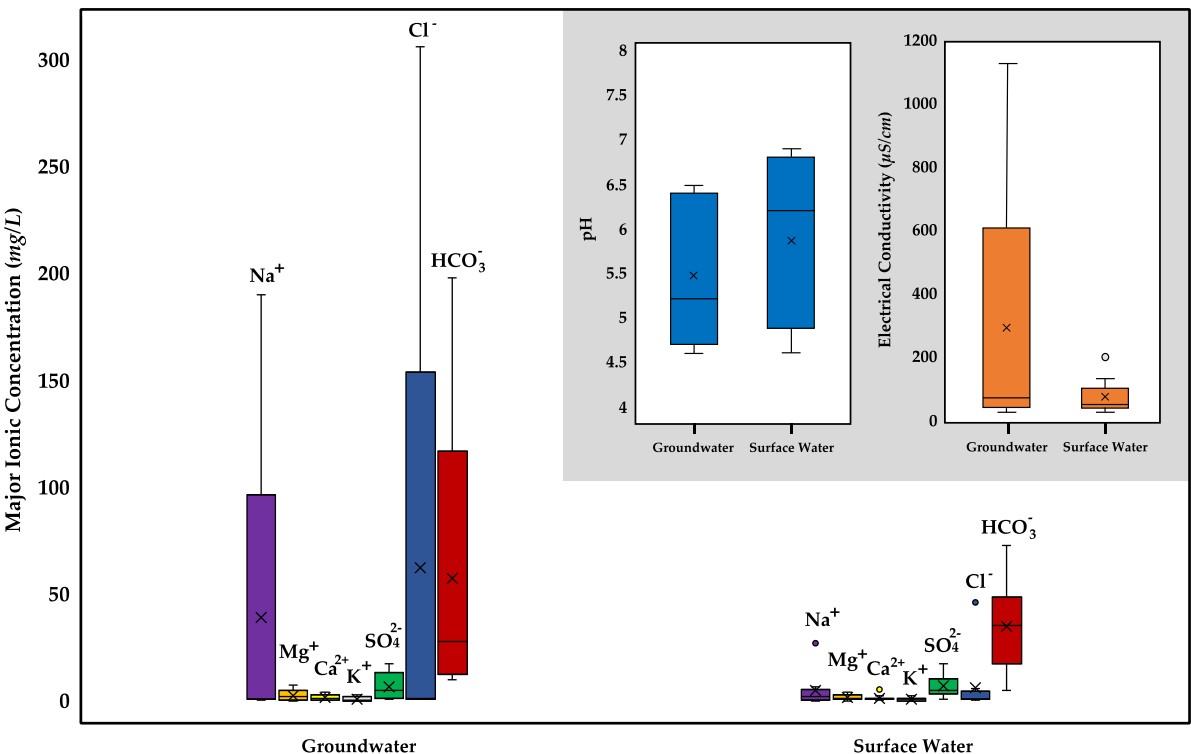

**Figure 3.** Hydrochemistry of ground and surface water (major ionic concentration); inset box plots showing the pH and EC values. Outliers are represented by 'o', and mean values are represented by '×'.

The pH results of samples in this study are slightly to moderately acidic, with pH values ranging from 4.6 to 6.5 for the groundwater samples and from 4.6 to 6.9 for the surface water samples. According to the FAO standard guideline, the normal pH range for irrigation water is 6.5 to 8.4 [16]. Irrigation water with pH outside this normal range may result in plant toxicity leading to nutritional imbalance in plants [60]. Therefore, pH values outside this range warrant further investigation and evaluation.

The EC of the analysed water samples in this study ranged from 28 to 1127 µS/cm for the groundwater samples and from 30 to 202 µS/cm for the surface water samples. According the FAO's permissible limits for irrigation use, EC values should be less than 750 µS/cm [16]. Higher EC values are indicative of a higher salt concentration in the water [50,61].

The ionic constituents of the analysed water samples indicate that the general abundance order for major cations is sodium > magnesium > calcium > potassium (Figure 3). Sodium ions vary from 0.7 to 190 mg/L for the groundwater samples and from 0.1 to 27 mg/L for the surface water samples. The results of magnesium ion concentration range from 0.2 to 7.4 mg/L for the groundwater samples and from 0.2 to 4.3 mg/L for the surface water samples. Calcium concentrations in the groundwater and surface water samples range between 0.4 to 4.3 mg/L and 0.5 to 5.5 mg/L, respectively. Among the analysed cations, potassium ions are the lowest in concentration, with values ranging from 0.1 to

2.8 mg/L for the groundwater samples and 0.1 to 2.4 mg/L for the surface water samples. Results show that all the measured cations in the water samples are lower than the FAO permissible limits, thus, suggesting their suitability for irrigation use.

The general order of abundance for major anions in the analysed water samples is bicarbonate > sulphate > chloride > carbonate (Figure 3). Bicarbonate is the most dominant anion, with values ranging from 10 to 198 mg/L for the groundwater samples and from 5 to 73 mg/L for the surface water samples. Results show that the bicarbonate concentrations of most of the analysed water samples are lower than the FAO's permissible limits (<150 mg/L), except for the groundwater sample G4 (198 mg/L) (Table 4). The range in concentrations of sulphate and chloride ions in the groundwater samples are 1 to 18 mg/L and 1 to 306 mg/L, respectively. In comparison, the range in the concentrations of sulphate and chloride ions in the surface water samples are 1 to 18 mg/L and 0.9 to 46 mg/L, respectively. Results showed that the sulphate and chloride concentrations in the analysed samples are lower than the FAO standard limits, suggesting their suitability for irrigation use. In addition, carbonate ions were absent in all the analysed samples.

### 3.1.2. Hydrochemical Facies

The major ionic constituents of the analysed water samples in this study were plotted on the Piper trilinear diagram for hydrochemical facies analysis [57]. Each facies type indicates the predominant cations and anions that affect the hydrochemistry of a sample [21,62]. The four major facies types represented in the Piper diagram are the calcium-, magnesium-, sulphate-, and chloride-type (Type I); sodium-, chloride-, and sulphate-type (Type II); sodium-, potassium-, and bicarbonate-type (Type III); and calcium-, magnesium-, and bicarbonate-type (Type IV). Results revealed that most of the analysed water samples belong to the 'Type IV' water category, two samples (S9 and G4) show a 'Type II' water category, one sample (S4) show a 'Type I' water category, and two samples (S10 and S7) show a 'Type III' water category (Figure 4).

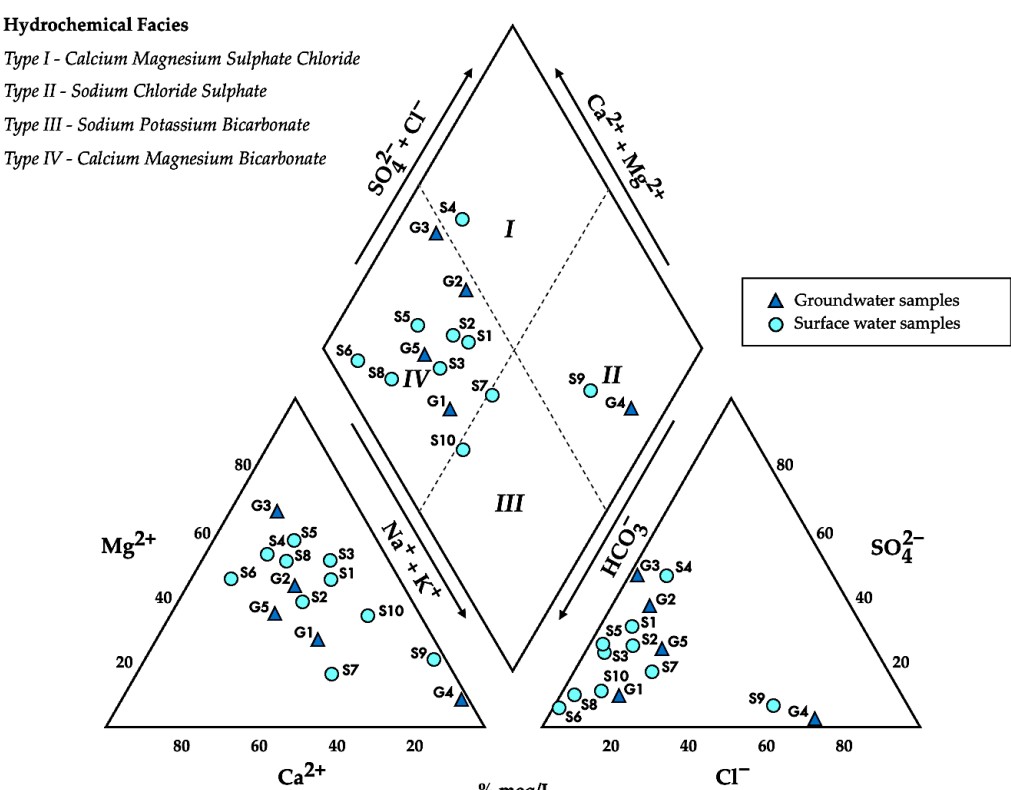

**Figure 4.** Piper trilinear diagram showing the hydrochemical facies of the analysed ground and surface water samples in the study area. Arrows are showing increasing amounts of the ions.

### 3.1.3. Heavy Metals Assessment

The measured concentration of heavy metals in the analysed ground and surface water samples are given in Table 5. The abundance order of measured metals in the analysed samples is Fe > Zn > Cu > Cr > As > Pb > Cd. Results showed that the measured heavy metal concentrations are within the FAO permissible limits and are deemed safe for irrigation use [16,62].

**Table 5.** Heavy metals concentration of ground and surface water samples in Brunei Darussalam.

| Sample | Sample ID | Fe | Zn | Pb | Cu | Cr | Cd | As |
|---|---|---|---|---|---|---|---|---|
| | | | | | mg/L | | | |
| Groundwater | G1 | 0.78 | 0.007 | <0.001 | <0.001 | <0.001 | <0.0005 | <0.001 |
| | G2 | 0.19 | 0.029 | <0.001 | <0.001 | <0.001 | <0.0005 | <0.001 |
| | G3 | 1.53 | 0.016 | <0.001 | <0.001 | <0.001 | <0.0005 | 0.002 |
| | G4 | 2.92 | 0.005 | <0.001 | <0.001 | <0.001 | <0.0005 | <0.001 |
| | G5 | 1.17 | 0.008 | <0.001 | <0.001 | <0.001 | <0.0005 | <0.001 |
| Surface Water | S1 | 0.94 | 0.018 | 0.0005 | 0.0065 | 0.0016 | <0.0001 | 0.0028 |
| | S2 | 0.73 | 0.012 | 0.0002 | <0.0001 | 0.0011 | <0.0001 | 0.0012 |
| | S3 | 0.10 | 0.009 | 0.0002 | <0.0001 | 0.001 | <0.0001 | 0.0012 |
| | S4 | 0.39 | 0.037 | <0.0001 | <0.0001 | 0.0009 | <0.0001 | 0.001 |
| | S5 | 0.10 | 0.041 | 0.0001 | <0.0001 | 0.0007 | <0.0001 | 0.0002 |
| | S7 | 0.64 | 0.051 | 0.0004 | <0.0001 | 0.001 | <0.0001 | 0.0009 |
| | S8 | 0.10 | 0.014 | <0.0001 | 0.0016 | 0.0007 | <0.0001 | 0.0004 |
| | S9 | 0.60 | 0.012 | 0.0002 | <0.0001 | 0.001 | <0.0001 | 0.0004 |
| | S10 | 0.09 | 0.017 | <0.0001 | <0.0001 | 0.0007 | <0.0001 | 0.0004 |
| | | | | Worldwide Standard Limits | | | | |
| FAO * | | <5.0 | <2.0 | <5.0 | <0.2 | <0.1 | <0.01 | <0.1 |

Note: * Standard permissible limits are those provided by the Food and Agriculture Organisation of the United Nations (FAO) [16].

### 3.2. Classification for Irrigation Suitability

Calculated water quality indices (SAR, Na%, RSC, MAR, KR, TH, PS, and IWQI) for classifying water suitability for irrigation use in this study are given in Tables 6 and 7. The general details of water quality indices are discussed in the sub-sections below:

**Table 6.** Irrigation water quality indices of ground and surface water samples in Brunei Darussalam.

| Sample | Sample ID | SAR | Na% | RSC | MAR | KR | TH | PS | IWQI |
|---|---|---|---|---|---|---|---|---|---|
| Groundwater | G1 | 0.2 | 42.6 | 0.22 | 45.2 | 0.64 | 1.5 | 0.07 | 3.1 |
| | G2 | 0.1 | 28.2 | 0.14 | 59.1 | 1.31 | 4.5 | 0.09 | 2.9 |
| | G3 | 0.1 | 13.0 | 0.36 | 74.8 | 0.32 | 14.7 | 0.22 | 5.8 |
| | G4 | 11.1 | 90.4 | 3.15 | 85.3 | 78.8 | 35.7 | 8.64 | 80 |
| | G5 | 0.3 | 27.8 | 0.31 | 46.9 | 0.43 | 20.2 | 0.12 | 7.7 |
| Surface Water | S1 | 0.4 | 36.1 | 0.42 | 69.2 | 0.61 | 20.2 | 0.29 | 9.1 |
| | S2 | 0.4 | 32.0 | 0.17 | 56.3 | 0.33 | 31.3 | 0.33 | 11.6 |
| | S3 | 0.3 | 34.7 | −0.40 | 75.9 | 0.14 | 11.2 | 0.07 | 4.5 |
| | S4 | 0.1 | 16.5 | −0.22 | 62.3 | 0.05 | 7.3 | 0.13 | 3.1 |
| | S5 | 0.1 | 21.3 | 0.07 | 72.8 | 0.06 | 7.1 | 0.11 | 5 |
| | S6 | 0.02 | 10.9 | 0.78 | 49.7 | 0.01 | 2.5 | 0.05 | 11 |
| | S7 | 0.2 | 52.9 | −0.55 | 32.0 | 0.04 | 2.6 | 0.04 | 2 |
| | S8 | 0.2 | 23.6 | −0.02 | 65.8 | 0.08 | 8.6 | 0.06 | 6.1 |
| | S9 | 2.4 | 76.4 | 0.56 | 85.0 | 4.41 | 14.8 | 1.36 | 16 |
| | S10 | 0.5 | 52.4 | 0.33 | 71.2 | 0.60 | 6.6 | 0.12 | 6.8 |

Notes: SAR: sodium adsorption ratio, RSC: residual sodium carbonate, MAR: magnesium adsorption ratio, KR: Kelly's ratio, TH: total hardness, PS: potential salinity, IWQI: Irrigation Water Quality Index.

### 3.2.1. Salinity Hazard

Electrical conductivity (EC) is the most crucial salinity hazard parameter in determining the suitability of water for irrigation use [50,61]. EC is the measure of the capacity of a substance to conduct electric current, which depends upon the temperature and salts present in the water [63,64]. Irrigation water with high salt levels may, in turn, increase the salt concentration in the soils, which can be an issue if the salt accumulates to a level harmful to crops [60]. Based on the classified EC values (Table 7), most of the analysed water samples in this study are generally safe for irrigation use. In contrast, the groundwater sample G4 showed a 'slight to moderate' salinity hazard (EC = 1127 µS/cm), suggesting restriction on the use for irrigation [16,50].

**Table 7.** Water classification based on the calculated water quality indices of ground and surface water samples in Brunei Darussalam.

| Indices | Range | Classification | Reference | Samples | |
|---|---|---|---|---|---|
| | | | | Groundwater | Surface Water |
| Electrical Conductivity (EC) | <750 | No problem | [16,50] | G1–G3, G5 | S1–S10 |
| | 750–3000 | Slight to Moderate | | G4 | - |
| | >3000 | Severe | | - | - |
| Sodium Adsorption Ratio (SAR) | <10 | Excellent | [16,50] | G1–G3, G5 | S1–S10 |
| | 10–18 | Good | | G4 | - |
| | 18–26 | Doubtful | | - | - |
| | >26 | Unsuitable | | - | - |
| Sodium Percentage (Na%) | <20 | Excellent | [64] | G3 | S4, S6 |
| | 20–40 | Good | | G2, G5 | S1–S3, S5, S8 |
| | 40–60 | Permissible | | G1 | S7, S10 |
| | 60–80 | Doubtful | | - | S9 |
| | >80 | Unsafe | | G4 | - |
| Residual Sodium Carbonate (RSC) | <1.25 | Good | [52] | G1–G3, G5 | S1–S10 |
| | 1.25–2.5 | Doubtful | | - | - |
| | >2.5 | Unsuitable | | G4 | - |
| Magnesium Adsorption Ratio (MAR) | <50% | Suitable | [53] | G1, G5 | S6, S7 |
| | >50% | Unsuitable | | G2–G4 | S1–S5, S8–S10 |
| Kelley's Ratio (KR) | <1 | Suitable | [54] | G1, G3, G5 | S1–S8, S10 |
| | >1 | Unsuitable | | G2, G4 | S9 |
| Total Hardness (TH) | <75 | Soft | [46] | G1–G5 | S1–S10 |
| | 75–150 | Moderately hard | | - | - |
| | 150–300 | Hard | | - | - |
| | >300 | Very hard | | - | - |
| Potential Salinity (PS) | <3 | Excellent | [51] | G1–G3, G5 | S1–S10 |
| | 3–5 | Good | | - | - |
| | >5 | Unsuitable | | G4 | - |
| Irrigation Water Quality Index (IWQI) | 0–25 | Excellent | [56] | G1–G3, G5 | S1–S10 |
| | 26–50 | Good | | - | - |
| | 51–75 | Poor | | - | - |
| | 76–100 | Very Poor | | G4 | - |
| | >100 | Unsuitable | | - | - |

### 3.2.2. Sodium Adsorption Ratio (SAR)

Sodium ions occur in water due to natural and anthropogenic processes [50]. However, high sodium levels are undesirable as they can lead to the development of alkaline soils. High sodium ions are more likely to be adsorbed onto clayey soils, replacing calcium and magnesium ions, resulting in poor soil permeability [65]. Therefore, the lower the sodium ions, the safer it is for plant growth [60]. The sodium adsorption ratio provides

a good estimate of sodium hazard in irrigation water [66,67]. SAR has been calculated as follows [50,68]:

$$SAR = \frac{Na^+}{\sqrt{\frac{Ca^{2+}+Mg^{2+}}{2}}}$$ (1)

where the sodium, calcium, and magnesium concentrations are expressed in meq/L.

SAR results revealed that all the analysed samples are classifiable as 'excellent' water class, except for groundwater sample G4 which is classifiable as a 'good' water class (SAR = 11) for irrigation use (Table 7). The analytical data were also plotted on the US salinity laboratory diagram [58] in Figure 5. The diagram is based on the integrated effects of EC and SAR. EC is representative of the salinity hazard (horizontal axis), and SAR is representative of the sodicity hazard (vertical axis). Each axis is separated into four hazard levels which are low, medium, high, and very high [65]. The application of the USSL diagram showed that most of the water samples are distributed within the C1S1 group, representing waters with low salinity and low sodium hazards, whereas one groundwater sample (G4) is categorised as C3S3, showing high salinity and sodium hazards (Figure 5).

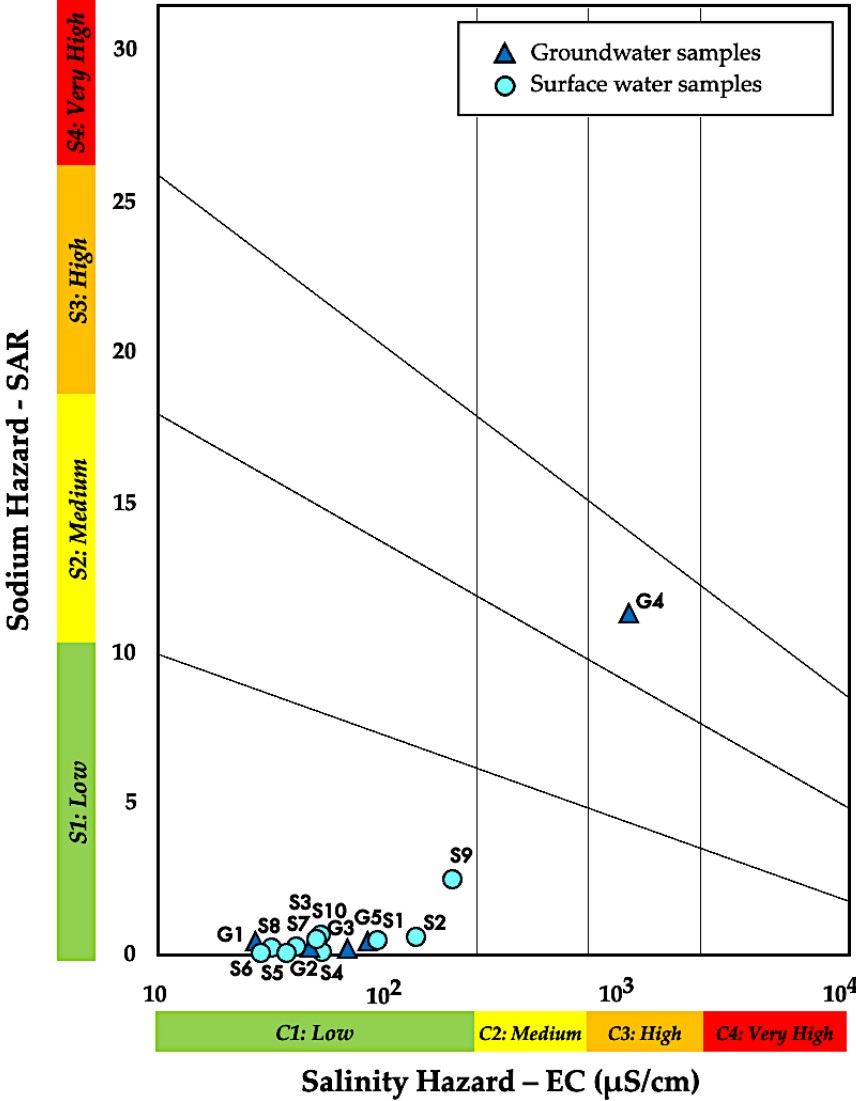

**Figure 5.** USSL diagram showing the irrigation suitability of the analysed ground and surface water samples in the study area.

### 3.2.3. Sodium Percentage (Na%)

Sodium percentage, or soluble sodium concentration, is used to evaluate the sodicity hazard of irrigation water [64,66]. Na% is calculated using the expression below [51]:

$$Na\% = \left( \frac{Na^+ + K^+}{Ca^{2+} + Mg^{2+} + Na^+ + K^+} \right) \times 100 \quad (2)$$

where the ionic concentrations are expressed in meq/L.

Based on the calculated Na% values of samples in this study (Tables 6 and 7), the majority of the samples demonstrate 'excellent to permissible' water quality (Na% lower than 60%). One sample (S9) plotted in the 'doubtful' water quality category (Na% = 76%), and one sample (G4) falls within the 'unsafe' water quality category (Na% = 90%). Analytical data were plotted on the Wilcox diagram [59] in Figure 6. The Wilcox diagram is based on the integrated effects of the EC and Na% indicators to classify water quality for irrigation use [66]. Results show that almost all the samples fall within the 'excellent to good' water quality category, except for the groundwater sample G4, which has been classified in the 'doubtful to unsuitable' water quality category (Figure 6).

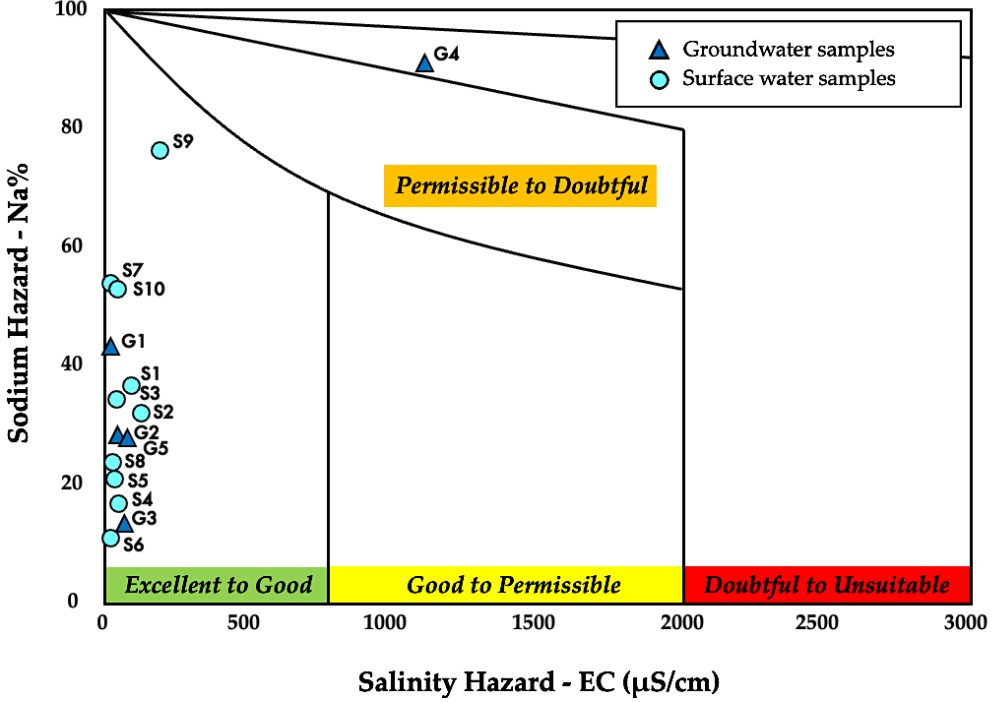

**Figure 6.** Wilcox diagram showing the irrigation suitability of the analysed ground and surface water samples in the study area.

### 3.2.4. Residual Sodium Carbonate (RSC)

Residual sodium carbonate is the difference between the total carbonate and bicarbonate ions minus that of alkaline earth elements (calcium and magnesium) [52,53]. Excess carbonate and bicarbonate ions (high RSC index) indicate that calcium and magnesium ions precipitate from the solution, resulting in higher sodium concentrations, thus increasing the potential sodium hazard [65,69]. The residual sodium carbonate (RSC) index is expressed as follows [52]:

$$RSC = \left( CO_3^{2-} + HCO_3^- \right) - \left( Ca^{2+} + Mg^{2+} \right) \quad (3)$$

where the ion concentrations are expressed in meq/L.

The RSC values of all of the analysed ground and surface water samples are representative of the 'good' water category (RSC < 1.25 meq/L), except for sample G4 (RSC = 3.15 meq/L), which can be classified as unsuitable for irrigation use (Tables 6 and 7).

### 3.2.5. Magnesium Adsorption Ratio (MAR)

The magnesium adsorption ratio index indicates the magnesium hazard of irrigation water [70]. High magnesium levels in irrigated soils are usually caused by an ionic exchange with sodium ions, resulting in poor soil permeability [64]. MAR ratios higher than 50% are deemed harmful and unsuitable for irrigation [64,70]. MAR is calculated as follows [53]:

$$\text{MAR} = \left( \frac{\text{Mg}^{2+}}{\text{Ca}^{2+} + \text{Mg}^{2+}} \right) \times 100 \tag{4}$$

where the calcium and magnesium ions are expressed in meq/L.

MAR values of the analysed ground and surface water samples in this study ranged between 32% to 85% (Table 6). Results showed that only 27% of the samples (G1, G5, S6 and S7) are classified as 'suitable' water for irrigation use, while others with MAR values higher than 50% are classified as 'unsuitable' for irrigation use (Table 7).

### 3.2.6. Kelley's Ratio (KR)

Kelley's ratio is an essential water quality index for determining water suitability for irrigation use [70]. Irrigation water quality is considered suitable if the KR is less than one and unsuitable if the KR is more than one. KR is expressed as follows [54]:

$$\text{KR} = \frac{\text{Na}^+}{\text{Ca}^{2+} + \text{Mg}^{2+}} \tag{5}$$

where the sodium, calcium, and magnesium concentrations are expressed in meq/L. KR values of the analysed water samples in this study varied from 0.01 to 78 meq/L (Table 6). Results show that most of the analysed samples are classified as 'suitable' water, except for the groundwater samples G2 and G4 and surface water sample S9, which are classified as 'unsuitable' waters for irrigation use (Table 7).

### 3.2.7. Total Hardness (TH)

Total hardness has been estimated from calcium and magnesium ions using the following equation and is calculated as mg/L CaCO$_3$ [46,63]:

$$\text{TH} = 2.497 \, \text{Ca}^{2+} + 4.118 \, \text{Mg}^{2+} \tag{6}$$

The TH of water can be classified into 'soft' or 'hard'. TH of less than 75 mg/L is for 'soft' water, 75 to 150 mg/L is for 'moderately hard' water, 150 to 300 mg/L is for 'hard' water, and more than 300 mg/L is for 'very hard' water [19,70]. The TH values of water samples in this study varied from 1 to 36 mg/L (Table 6). Results show that all the analysed samples in this study are classified as 'soft' water (Table 7).

### 3.2.8. Potential Salinity (PS)

The PS index is used for classifying irrigation water based on chloride ion concentrations and half of the sulphate ion concentrations. PS values lower than five are deemed suitable for irrigation use [69]. PS is calculated using the formula [51]:

$$\text{PS} = \text{Cl}^- + \frac{1}{2}\text{SO}_4^{2-} \tag{7}$$

The ionic concentrations of chloride and sulphate are expressed in meq/L.

In this study, the value of PS varied from 0.06 to 8.6 meq/L (Table 6). Results suggest that most of the analysed water samples fall within the 'excellent' water class, except for

the groundwater sample G4 (PS = 8.6 meq/L), which can be classified as 'unsuitable' water for irrigation use (Table 7).

3.2.9. Irrigation Water Quality Index (IWQI)

The IWQI is a single dimensionless number used to evaluate the health or condition of a water body through integrating multiple water quality parameters, providing a comprehensive assessment of water quality in a simplified manner [24,71]. In the present study, the irrigation water quality index was calculated based on the standard three-step methodology [56].

First, specific parameters were identified and assigned a weight ($w_i$) according to the relative importance to the overall water quality [69]. The specific parameters used for calculating the IWQI in the present study are EC, sodium, chloride, bicarbonate, and SAR [72,73]. For each of the chosen parameters, the standard limit is prescribed by the FAO [16]. The relative weight ($W_n$) was calculated for each parameter based on the following equation [56]:

$$W_n = \frac{k}{S_n} \tag{8}$$

where $W_n$ is the unit weight factors for the nth parameter and $S_n$ is the standard desirable value of the nth parameter. k is the proportionality constant and was calculated using the following formula:

$$k = \frac{1}{\sum \frac{1}{S_n}} \tag{9}$$

On summation of all selected parameters' unit weight factors, $W_n = 1$.
Secondly, the sub-index value ($Q_n$) was calculated as follows:

$$Q_n = \frac{[(V_n - V_o)]}{[(S_n - V_o)]} \times 100 \tag{10}$$

where $Q_n$ is the sub-index value, $V_n$ is the mean concentration of the nth parameter, and $V_o$ is the actual values of the parameters in pure water.

Thirdly, through combining the first and second steps, the overall IWQI was calculated using the following formula:

$$IWQI = \frac{\sum W_n Q_n}{\sum W_n} \tag{11}$$

The IWQI values obtained from the analysed water samples in the study area varied from 8 to 80 (Table 6). Water classification based on IWQI revealed that most of the analysed samples are of 'excellent' water quality, except for the groundwater sample G4, which showed 'very poor' water quality (Table 7).

*3.3. The Impact of Water Quality on Agricultural Development*

Water quality has a significant impact on agricultural development as it directly affects crop growth and productivity [16,64]. Contaminants such as salts and heavy metals present in water can hinder nutrient uptake, impair plant health, and reduce crop yields [60]. Poor water quality can lead to soil degradation, nutrient imbalance, and equipment clogging in irrigation systems. Furthermore, water pollution from agricultural runoff could also degrade water quality in its path and harm ecosystems [74]. Therefore, good quality water is crucial for ultimately driving agricultural development towards a more productive and sustainable future.

In the present study, hydrochemical evaluations revealed that waters in the study area primarily belong to the calcium-, magnesium-, and bicarbonate-type, or Type IV water category. There was no clear distinction between the ground and surface water types (Figures 3 and 4). In general, the analysed groundwater samples can be considered more mineralised compared to the surface water samples, and their pH levels are also more acidic in comparison (Figure 3). Although most of the investigated water is considered safe for

irrigation use and is not highly mineralised based on the FAO standard, potential harmful effects to crops and soil may only show after years of using the water for irrigation [66]. Low-pH waters also require further treatments to avoid increasing the soil acidity in the long run [14]. Furthermore, assessments of heavy metals revealed that all the analysed water samples are deemed safe for irrigation use (Table 5). Iron is the most-detected heavy metal in all the analysed samples. In the study area, trace metals are notably higher in residential, agricultural, and industrial waterways [26]. Iron is also present in the sandstones of Brunei [75].

Our findings revealed mainly low salinity and sodicity risks in all the analysed samples as evident from the EC, SAR, Na%, RSC, KR, and PS values and classifications (Tables 6 and 7). However, for sample G4, high salinity and sodicity concerns have been observed. It is believed that groundwater contamination has occurred at the well site due to anthropogenic activities such as the overuse of fertilisers and infiltration of mineralised irrigation water into the aquifer [37]. Based on the magnesium hazard assessment (MAR), our findings showed that only 27% of the analysed water samples in this study are considered suitable for irrigation use. Therefore, implementation of appropriate water management practices in the study area will help mitigate the potential impacts of magnesium hazard in the irrigation water [14]. Limitations on the use of highly mineralised waters should be considered, preferably for salt-tolerant crops. A good drainage system is also required to further prevent soil salinisation [61,64].

The general water quality index (WQI) system has found significant application in Southeast Asia due to the region's abundant water resources and the need for reliable water management [23,76,77]. The application of WQI in Brunei Darussalam, however, has not been investigated before. In this study, we employed the irrigation water quality index (IWQI) to evaluate the water quality status of selected ground and surface water resources gathered from all four districts of Brunei. Results showed 'excellent' water quality for the majority of the analysed samples, except for the groundwater sample G4, which placed in the 'very poor' water class (Table 7). Poor water class based on the IWQI indicates that the water is polluted or deteriorated and is deemed unsuitable for irrigation use [56,69]. Regular water quality monitoring is therefore recommended in areas with poor MAR and IWQI results.

Future water quality studies in the country should focus on the impact of land use practices, such as deforestation for agriculture and urban development [78,79]. Addressing water quality issues in Brunei Darussalam also requires a comprehensive and integrated approach involving government agencies, industries, communities, and stakeholders [27,32]. An integrated approach for water quality monitoring through combining traditional sampling methods and advanced technologies, such as remote sensing, drones, and sensor networks, should be utilised [48,80]. Furthermore, the use of real-time data collection, data analytics, and modelling techniques could further enhance water quality monitoring efficiency, improve data accuracy, and facilitate timely decision-making.

## 4. Conclusions

Water quality evaluations for classifying water suitability for irrigation use, in particular, were conducted in Brunei Darussalam. Based on the measured pH, water samples in the study area are slightly to moderately acidic. Most of the samples are classified as Type IV water class: calcium-, magnesium-, and bicarbonate-type water. Our assessments of heavy metals (Fe, Zn, Cu, Cr, As, and Cd) revealed that the water samples are generally safe for irrigation use. Salinity and sodicity hazard assessments based on EC, SAR, Na%, KR, and PS showed that almost all the analysed water samples are classified as suitable for irrigation. However, magnesium hazards based on MAR were observed in most of the samples. Overall, the IWQI of water samples in this study showed 'excellent' water, except for sample G4 which showed 'very poor' water. It is therefore recommended to apply suitable treatment to these water resources before using them for irrigation, particularly for waters with poor MAR and IWQI results. Moreover, potentially harmful effects to soils

and crops may only show after years of using the water for irrigation. Our findings show that the water resources in Brunei Darussalam are stressed and, to some degree, impacted by natural and anthropogenic causes. Future studies should include regular water quality monitoring to ensure sustainable management of water resources in the country.

**Author Contributions:** Conceptualisation, S.L.A. and S.H.G.; methodology, S.L.A. and C.S.T.; validation, S.L.A., C.S.T., L.H.L. and S.H.G.; formal analysis, S.L.A., C.S.T. and M.S.; investigation, S.L.A. and S.H.G.; resources, L.H.L., M.F.I., M.S. and S.H.G.; data curation, S.L.A. and C.S.T.; writing— original draft preparation, S.L.A.; writing—review and editing, L.H.L., M.S. and S.H.G.; visualisation, S.L.A.; supervision, L.H.L. and S.H.G.; project administration, S.H.G.; funding acquisition, S.L.A., M.S. and S.H.G. All authors have read and agreed to the published version of the manuscript.

**Funding:** This research was funded by the Coordinating Committee for Geoscience Programmes in East and Southeast Asia and the Geological Society of America through the East Asian Geoscience and Environmental Research (EAGER) project, Maurice 'Ric' Terman award 2021, grant number UBD/RSCH/1.21.3[a]/2021/011.

**Data Availability Statement:** The datasets used in this study are available from the corresponding author (S.L.A.) upon reasonable request.

**Acknowledgments:** The authors thank the Ministry of Primary Resources and Tourism, Department of Agriculture and Agrifood, Brunei Darussalam, for valuable discussions on the sampling locations. The authors are highly grateful to the National Committee of Geosciences Brunei (NCGeo) for their support and assistance during the grant application. Badri Suhaili is thanked for field assistance. The authors thank Britt Reto for analysing samples at the Eawag laboratory. The authors further acknowledge the service and laboratory support from Qasya Diagnostic Services (QLAS) Sdn Bhd.

**Conflicts of Interest:** The authors declare no conflict of interest.

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
