# Peer review of "Evaluation of Ground and Surface Water Hydrochemistry for Irrigation Suitability in Borneo: Insights from Brunei Darussalam"

_water, doi:10.3390/w15122154_

Round 1

Reviewer 1 Report

The topic of this manuscript falls within the scope of the journal. The abstract should be written to attract the journal’s audience. The Introduction and Discussion should be completed with recent articles on the same topic published in international peer-reviewed journals. The major problem of this manuscript is that the data presented needs to be quality assured and controlled. The analytical methods' detection limits and quality control should be reported.  Additional comments are included in the attached .pdf file.

Regards

Please see the attached .pdf file.

Reviewer 2 Report

Comments to the authors:

The authors performed an evaluation of ground and surface water hydrochemistry for irrigation suitability in Borneo, Brunei Darussalam. Overall, the study is methodological sound, with promising results and discussion. However, prior to further consideration, some comments need to be addressed.

(i) Title: I would suggest removing A case study from

(ii) Introduction: Huge modifications are necessary for the introduction section. The research gaps and significance of the study are not shown. Moreover, the authors should provide more precise research objectives in this study.

(iii) Why no water quality index was used? It is the most commonly used classification method to evaluate the water quality, including different parameters, giving a comprehensive evaluation. Do include it for further analysis.

(iv) Why did the authors use FAO irrigation standard? Does Brunei has any own national sandard?

(v) What is the monitoring frequency and duration of the monitoring parameters?

(vi) Discussion is lacking in this study. So what are the proposed mitigations? What should the government do for proper water quality management? Effects from changing land use or anthropogenic activities? The difference between upstream or downstream is not provided.

The suggested papers are meant to improve the authors' work for enhancing current work. Sure. Here are few more papers which j think can help to improve the introduction section of the manuscript.

a.      doi:10.1007/s10661-020-08543-4.

b.      doi: 10.1007/s10661-021-09202-y

c.     Water Quality Assessment and Monitoring in Pakistan: A Comprehensive Review

d.  Integration of advanced optimization algorithms into least-square support vector machine (LSSVM) for water quality index prediction

e.     A comprehensive review of water quality indices (WQIs): history, models, attempts and perspectives  

(viii).  Conclusion section seems to be a repetition of the results section. Huge modifications are required. Please provide insights into this study and what can be further done in the future.

(ix).  Practical and Theoretical Implications for future research may also be included in the conclusion at the end.

Moderate English Revisionis Required.

Reviewer 3 Report

Very nice, descriptive paper on water resource chemistry of Brunei-Darussalam. Well-presented and logically written. A bit formulaic, but this is good for a case study.

I think it worthy of publication and have 3 comments that I would like addressed:

1. The abstract requires an additional sentence to explain why this work was done. As far as I can glean, this is due to data paucity and the increasing use of both surface and groundwater without appreciation of potential hydrochemical impacts and limitations, especially if groundwater is to be increasingly used as an adjunct to surface water supplies.

2. The discussion needs additional recommendation to assess other potential irrigation contaminants, such as nutrients, pathogens, faecal coliforms, etc.

3. Groundwater sample G4 is quite different, but there is only a passing reference to possible reasons ("believed to be a result 354 of anthropogenic activities such as overuse of fertilisers and surface irrigation methods" l.355). I note that this was the only non-artesian bore and had to be flushed prior to sampling. In my experience, 25 minutes is inadequate to develop a representative sample. Rather, flushing should continue until physico-chemical parameters have stabilised, or more than 3 casing volumes, whichever takes longer. This needs to be highlighted and can reasonably explain the anomalous results. Alternatively, you should explain why you believe the concentrations measured are a true representation of the groundwater in this area.

Reviewer 4 Report

Dear authors,

i believe that there is alot of work that has been done.

There are a few things that must be improved in order to get this one published..

One other significant drawback is the methodology. It is not supported by references adequately.

The conclusions aren't supported by the results and there is a clear lack of future research and reference to the overall merit of this research paper.

Best,

Please check the language. There are some sentences that doesn't make a clear sense..

Reviewer 5 Report

Dear Authors, I found your work interesting and important. However, your research article seems like a report (rather than a scientific paper). Please find below my comments and suggestions:

1. The water quality sample data is collected and analzed in laboratory and the results were presented. There is no discussion with respect to the previous studies.

2. You must include a research component which could be exploring the link between variation in water quality with respect to land use and  geology. This is completely missing.

3. What is innovative about your research? Please indicate in your article. 

4.  Section 2.1.3. is not a part of materials and methods section. This, should be moved to literature review. 

5. Why only 15 sampling locations (10 surface and 5 ground water) were considered? Are the samples sufficient to represent the case study? Please justfiy. 

OK

Round 2

Reviewer 1 Report

The authors have to carefully check again my initial comments and further improve the manuscript. The Introduction and Discussion sections should be improved.

Regards,

The quality of grammar and syntax is improved in this version of the manuscript.

Reviewer 2 Report

The authors have substantially addressed my comments during the first review. However, there are some minor comments to be addressed:

(i) Table 1 and Table 2: Characterization should be characteristics

(ii) Section 2.3: Groundwater Classification

(iii) Relevant references for WQI can be included (Lines 415-434):

(a) DOI: 10.1007/s10661-021-09202-y

(b) https://doi.org/10.2166/ws.2021.303

(iv) Major revision is required in section 3.3: Some sentences are repeated and not necessary. For instance Lines 445-446 and 495-496. Reduce the redundancy and provide precise discussion. Too lengthy and has no focal point. Some of the points can be put into the conclusion section.

Proofreading is necessary. 

Reviewer 4 Report

 I believe you have met the requirements.

The quality of the English language should be improved..

Round 3

Reviewer 1 Report

The manuscript has reached the level of acceptance for publication in Water.

Regards,

The manuscript has reached the level of acceptance for publication in Water.

Regards,